# Mapping Extracted Free-Text Primary Diagnoses to ICD-10 and SNOMED-CT Using SciSpacy - A Performance Evaluation

Parvati Naliyatthaliyazchayil
*Department of Biomedical Informatics and Engineering*
*Indiana University*
Indianapolis, USA

Venkat Ramana Sangam
*Department of Biomedical Informatics and Engineering*
*Indiana University*
Indianapolis, USA

Raajitha Muthyala
*Department of Biomedical Informatics and Engineering*
*Indiana University*
Indianapolis, USA

Saptarshi Purkayastha
*Department of Biomedical Informatics and Engineering*
*Indiana University*
Indianapolis, USA

*Abstract*—Accurate extraction and standardization of clinical diagnoses from unstructured electronic health records (EHRs) remain a critical challenge in healthcare data science. This study is among the first to evaluate the performance of scispaCy for mapping diagnosis concepts extracted from MIMIC-IV clinical notes to standardized medical codes. Our natural language processing (NLP) pipeline leverages scispaCy to map diagnosis concepts to Unified Medical Language System (UMLS) Concept Unique Identifiers (CUIs) and crosswalk these to ICD-10 and SNOMED-CT codes. Applied to the MIMIC-IV dataset, the pipeline demonstrated robust coverage, successfully mapping 94.1% of extracted diagnosis concepts to UMLS CUIs across 98% of patients, with 80.3% of patients having CUIs mapped to ICD-10. Exact ICD-10 code matches between model output and MIMIC-IV diagnosis records were observed in 58.3% of patients, while a hierarchical category level roll-up comparison improved matching to 83.1%, reflecting clinical coding complexities. The pipeline's reliance on UMLS CUIs offers versatility across coding standards, and its design supports integration with existing EHR systems using standard hardware, enhancing accessibility. Our approach poses a lower risk of hallucination and reduces gender and racial bias compared to large language models, as it relies on structured vocabularies rather than generative deep learning. This work highlights the promise of combining rule-based and statistical NLP methods for scalable, transparent, and clinically relevant diagnosis mapping, with the potential to improve research applications and clinical decision support systems.

*Index Terms*—Natural language processing, scispaCy, ICD-10, SNOMED, MIMIC-IV, Clinical coding, Diagnosis mapping, Medical concept extraction

## I. INTRODUCTION

Electronic Health Records (EHRs) have become foundational to modern healthcare, offering a digital infrastructure for storing both structured and unstructured patient data [1]. Structured data refers to standardized, pre-defined fields such as lab results, medication codes, and vital signs, etc., organized in relational formats suitable for computation. In contrast, unstructured data consists of narrative, free-text entries like clinical notes, discharge summaries, and diagnosis descriptions, which lack a consistent format and are more difficult to process automatically [2,3]. While structured data benefits from established standards that ensure semantic interoperability and machine-readability [4], unstructured clinical text, despite containing richer and more nuanced insights, presents challenges due to its inconsistent formatting, lack of standardization, and variable data quality [5,6].

Historically underutilized, unstructured clinical text is gaining renewed attention due to advances in Natural Language Processing (NLP) and biomedical informatics. These advances have made it increasingly feasible to extract and structure meaningful information from narrative content [5,7]. Studies have shown that unstructured data often captures ambiguity, missingness, and clinical subtleties that structured data may overlook [7]. This has important implications for secondary data use and diagnostic decision-making. In the context of clinical trials, unstructured text often contains critical information such as eligibility criteria for patients, details of adverse events, and nuanced results that are not routinely captured in structured fields, thus playing a pivotal role in cohort identification and trial monitoring [7]. To enhance interoperability and computational utility, there is growing interest in mapping free-text diagnoses to standardized vocabularies such as the International Classification of Diseases (ICD-10) and the Systematized Nomenclature of Medicine - Clinical Terms (SNOMED-CT).

A range of tools and methods has emerged to support this codification process, spanning rule-based systems to deep learning-based classifiers. Dong et al. (2022) highlighted key challenges in developing robust and explainable clinical coding

systems, particularly around generalization, zero-shot code prediction in Machine Learning models, and integration of domain knowledge [8]. Other approaches, such as those by Silva H et al. (2024), have employed cosine similarity for ICD-10 code suggestion [9], while models like CODER use cross-lingual contrastive learning to capture semantic similarity in concept embeddings [10], although they are not directly optimized for codification. Abdulnazar et al. (2024) proposed an unsupervised bi-encoder model for SNOMED-CT annotation, showing promise for pre-labeling tasks in low-resource settings but notes ongoing challenges with contextual interpretation, false positives, and clinical language variability, suggesting the need for further refinement [11].

Among these, scispaCy has emerged as a widely used general-purpose biomedical NLP toolkit, offering pre-trained models for tasks such as named entity recognition (NER) and entity linking [12,13]. Importantly, scispaCy's entity linking models are trained to map extracted entities to concepts in the Unified Medical Language System (UMLS) [14] a comprehensive biomedical thesaurus that integrates a wide range of standard clinical terminologies, including ICD-10, SNOMED-CT, LOINC, and others. This connection positions scispaCy as a theoretically promising tool for clinical codification, as it can inherently leverage semantic interconnections between medical vocabularies through UMLS.

While recent advances in Large Language Models (LLMs) have demonstrated few-shot and zero-shot performance on a variety of clinical NLP tasks, these models are often computationally intensive and not yet optimized for clinical mapping [15]. In contrast, lightweight, domain-specific tools like scispaCy offer a more transparent and resource-efficient alternative. However, to date, no studies have systematically evaluated scispaCy's effectiveness for mapping to ICD-10 or SNOMED-CT using real-world datasets such as Medical Information Mart for Intensive Care (MIMIC-IV) [16], despite its widespread use for the extraction of biomedical entities. Such an evaluation is critical to understanding the practical utility and limitations of off-the-shelf biomedical NLP tools in supporting clinical codification workflows.

To address this gap, we propose and evaluate a novel two-stage pipeline for diagnosis code mapping using scispaCy. In the first stage, primary diagnosis statements are extracted from MIMIC-IV clinical notes using a rule-based method developed for this purpose. In the second stage, these extracted diagnoses are processed using scispaCy to identify biomedical entities and map them to ICD-10 and SNOMED-CT codes. We evaluated the performance and practical implications of the system, especially its potential for integration into clinical coding workflows.

To our knowledge, this is the first systematic evaluation of scispaCy for diagnosis code mapping using MIMIC-IV discharge summaries. This work contributes practical insights into the performance of open-source biomedical NLP tools and highlights their potential, particularly in a low-resource setting.

## II. METHODOLOGY

We outline below the detailed methodology employed in constructing and assessing the proposed pipeline.

### A. Dataset Description

We utilized the Medical Information Mart for Intensive Care IV (MIMIC-IV) dataset, a publicly available, de-identified database [17] containing comprehensive clinical data from patients admitted to critical care units at the Beth Israel Deaconess Medical Center between 2008 and 2019. Specifically, we focused on the MIMIC-IV Note module, which includes unstructured clinical documentation such as discharge summaries, progress notes, and nursing observations.

For this study, we leveraged discharge summaries, as they routinely contain definitive primary diagnostic statements that reflect clinicians' final assessments at the end of a hospital stay. Our analysis encompassed all discharge summaries associated with unique subject IDs in the dataset, resulting in a corpus of over 145,000 patients and their corresponding diagnosis narratives.

### B. Primary Diagnosis Extraction and Cleaning

To isolate relevant diagnosis content, we developed an automated rule-based script to extract text from sections of the discharge summaries explicitly annotated as "Discharge Diagnosis" and "Primary Diagnosis" by MIMIC-IV [17]. Within these targeted segments, we further processed the text to identify individual diagnosis entities. Multi-condition spans were parsed into separate entries by applying syntactic splitting rules, removal of extraneous tokens (e.g., punctuation, stopwords, section labels), deduplicating concepts, and formatting standardization was done as part of cleaning [18]. This preprocessing enabled the scalable extraction of over 1 million diagnosis entries across the cohort. To validate extraction accuracy, we manually reviewed a random sample of 200 diagnosis statements, confirming the precision of the heuristic method of extraction and cleaning. All processing was performed in Python, and the codebase is publicly available on GitHub - https://github.com/pnaliyatthaliyazchayil/scispacy_pipeline_analysis.

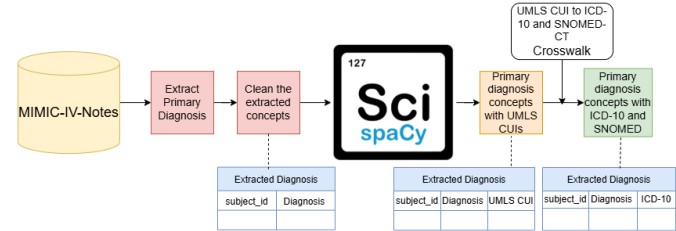

Fig. 1.  Free-text Diagnosis Mapping Pipeline using SciSpacy

### C. Mapping Concepts to UMLS Concept Unique Identifier Using scispaCy

Each cleaned diagnosis statement was individually processed using scispaCy, an open-source Python library developed by the Allen Institute for AI for biomedical text

processing [13]. Built on top of spaCy, scispaCy provides pre-trained models optimized for biomedical language tasks, including named entity recognition (NER) and entity linking to UMLS [14]. For this task, we employed the 'en_core_sci_lg' model, which includes a large vocabulary, domain-specific word embeddings, and an entity linker trained on UMLS[19]. A summary of the end-to-end pipeline is presented in Figure 1.

For each row, this module follows these steps:

- NER Step: The diagnosis text is tokenized and parsed to identify one or more biomedical entities.
- Entity Linking Step: Each identified entity is matched against UMLS concepts using a string similarity algorithm, returning the best candidate Concept Unique Identifier (CUI) along with metadata such as similarity score.

This process results in a list of recognized biomedical entities for each diagnosis, each mapped to one or more UMLS CUIs which serve as standardized representations. The output of this module is a structured dataset linking free-text diagnosis inputs to their corresponding UMLS identifiers.

### D. Crosswalking UMLS CUIs to ICD-10 and SNOMED-CT Codes

After assigning UMLS CUIs to the extracted diagnosis terms, we crosswalked them to ICD-10 and SNOMED-CT codes using the UMLS, specifically the MRCONSO.RRF file, which links CUIs to standardized vocabularies [20]. ICD-10 is primarily used for billing, reporting, and epidemiological surveillance [21], while SNOMED-CT provides a clinically rich, hierarchically structured terminology suited for documentation, interoperability, and clinical decision support [22].

Since concept CUIs can be associated with both code systems, we employed a hierarchical mapping strategy: ICD-10 codes were prioritized when available. Otherwise, SNOMED-CT codes were selected. To improve semantic accuracy, we restricted mappings to preferred terms (term type: 'PT') within each vocabulary. This strategy improved coding coverage while preserving alignment with the diagnostic focus of our use case. The choice to prioritize ICD-10 codes over SNOMED-CT was driven by the predominance of ICD-10 coding in the MIMIC-IV database. This facilitates direct comparison of the scispaCy-generated outputs with the database's ground truth.

### E. Evaluation

To assess the performance of our mapping pipeline, we compared the ICD-10 codes generated by the system against the ground truth ICD-10 codes in the MIMIC-IV dataset. We conducted two types of evaluations: an exact code comparison and a roll-up comparison to the category level parent code. Since clinically relevant predictions can be as important as exact matches, we included a category-level roll-up, where both predicted and ground truth ICD-10 codes were rolled up to their first three characters. This level of abstraction corresponds to the diagnostic category defined by the World Health

Organization (WHO) in the ICD-10 hierarchy [23] and is widely used in clinical and epidemiological research to group related conditions [24]. For example, I25.11 (Atherosclerotic heart disease of native coronary artery with angina pectoris) and I25.1 both fall under the broader category I25 (Chronic ischemic heart disease) and are rolled up to it, as illustrated in Figure 2. By evaluating performance at this level, we allow partial credit for predictions that capture the correct clinical condition even when the specific subcode differs. This approach acknowledges variability in subcode assignment and emphasizes clinically meaningful alignment.

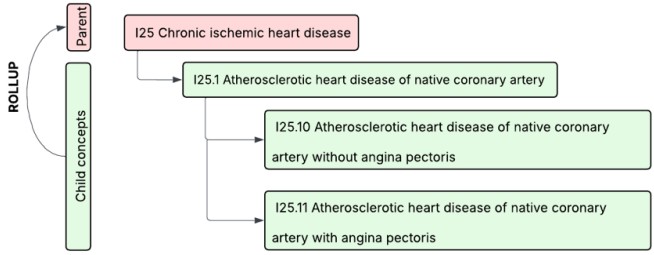

Fig. 2. Example of ICD-10 roll-up to parent level in ground truth and pipeline-generated codes for evaluation of clinical relevance

## III. RESULTS

### A. Evaluation of Extracting Diagnosis Concepts

The MIMIC-IV dataset includes Diagnosis records for 221,122 patients. Following automated extraction and cleaning of "Discharge Diagnosis" and "Primary Diagnosis" sections from discharge summaries, we obtained 1,078,656 cleaned diagnosis concepts associated with 145,219 patients. As expected, individual patients may have multiple diagnoses across different encounters. Some subject records did not contain the targeted diagnosis sections within their discharge summaries, and therefore, it is expected that the number of patients represented in the extracted data is lower than the total number available in the MIMIC-IV Diagnosis.

### B. Evaluation of Mapping Diagnosis Concepts to UMLS CUIs

Using SciSpaCy, we successfully mapped 1,015,591 (94.1%) of the extracted diagnosis concepts to UMLS CUIs, covering 142,468 out of the 145,219 patients (98.1%) included in the diagnosis extraction step. The remaining 1.9% of patients could not be assigned CUIs due to ambiguous phrasing of diagnosis terms, limitations in the vocabulary, or unmatched terminology. An overview of mapping coverage is presented in Figure. 3.

### C. Evaluation of Crosswalking UMLS CUIs to ICD-10 and SNOMED-CT Codes

After mapping diagnosis concepts to UMLS CUIs, we crosswalked each CUI to standardized clinical terminologies, ICD-10 and SNOMED-CT. Out of the 142,468 patients with mapped CUIs:

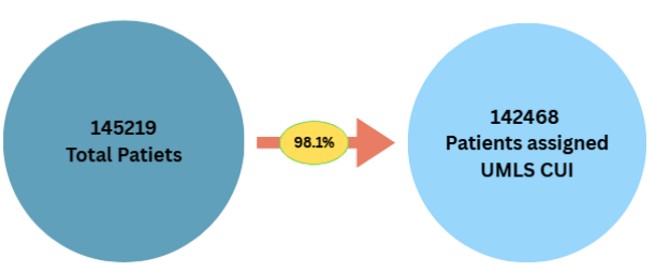

Fig. 3. Patients with UMLS CUIs mapped

- 114,501 patients (80.3%) had at least one diagnosis concept mapped to an ICD-10 code.
- 23,933 patients (16.8%) had diagnosis concepts mapped to SNOMED-CT codes.

These mapping counts are mutually exclusive due to our hierarchical mapping strategy, which prioritizes ICD-10 codes when a concept is linked to both terminologies. Figure. 4 provides a detailed breakdown of both the number of diagnosis concepts and unique patients successfully mapped to ICD-10 and SNOMED-CT codes, illustrating coverage from the perspectives of extracted concepts and patient representation.

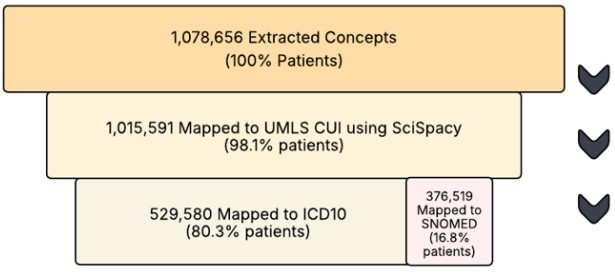

Fig. 4. Number of diagnosis concepts and unique patients successfully mapped to ICD-10 and SNOMED-CT codes

### D. Comparison of Model-Generated ICD-10 Codes to MIMIC-IV Ground Truth

To evaluate the accuracy of our NLP-generated diagnosis codes, we compared them against the MIMIC-IV Diagnosis codes. The MIMIC-IV Diagnosis table primarily contains ICD-10 codes, with a small subset of ICD-9 codes. To ensure direct comparability, we first crosswalked all ICD-9 codes to their corresponding ICD-10 equivalents prior to comparison.

Following this standardization, we joined the model-generated Subject_IDs with ICD-10 codes to the ground truth Subject_IDs from the MIMIC-IV Diagnosis table (post-crosswalk). Only patients with ICD-10 codes present in both the model output and the ground truth were retained for evaluation, resulting in a final comparison cohort of 102,539 patients.

We then conducted two types of comparisons against the MIMIC-IV ICD-10 ground truth codes:

- Direct Code-Level Comparison: We directly compared the generated ICD-10 codes with the exact ICD-10 codes documented for each patient in MIMIC-IV. This strict matching assesses the exact code-level accuracy of the model output. Out of the 102,539 patients in the evaluation cohort, 59,742 patients (58.3%) had at least one model-generated ICD-10 code that exactly matched a ground truth code.
- Hierarchical Roll-up Comparison: Recognizing that some ICD-10 codes vary only in their level of specificity (parent vs. child codes), we also performed a roll-up comparison. Both generated and MIMIC-IV codes were rolled up to their category level parent codes within the ICD-10 hierarchy, as shown in an example, Figure 2. Accuracy was then evaluated by comparing these parent-level codes, allowing partial credit when the model predicts a clinically relevant broader category, even if the precise subcode is missed. Out of the 102,539 patients in the evaluation cohort, 85,185 patients (83.1%) had at least one model-generated ICD-10 code that matched a ground truth code at the parent roll-up level.

This dual evaluation approach balances strict code-level precision with clinically meaningful hierarchical correctness, providing a nuanced understanding of the model's performance. These two were independent, parallel comparisons, both assessing model-generated ICD-10 codes against the MIMIC-IV ground truth for the same cohort of 102,539 patients. The evaluations were conducted separately and are not mutually exclusive, and are shown in Figure 5.

For external benchmarking, a recent study evaluated the out-of-the-box performance of several LLMs for ICD coding. The best-performing LLM (OpenAI O3) achieved 45.3% exact-code accuracy [25], notably lower than the 58.3% exact-code accuracy (83.1% at the category level) achieved by this scispaCy-based pipeline.

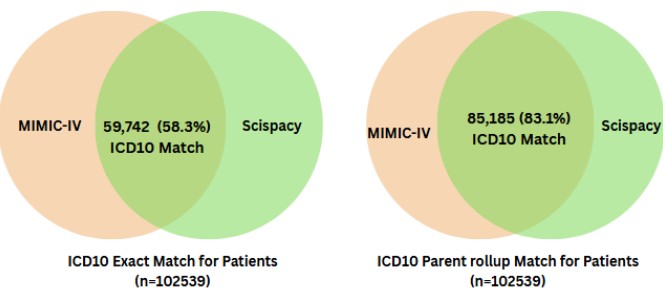

Fig. 5. Comparison of ICD10 codes generated by scispaCy to MIMIC-IV

### E. F1 score for Model Generated ICD-10 Codes

The F1 score is a harmonic mean of precision and recall, commonly used to evaluate classification tasks by balancing false positives and false negatives. In our case, ICD-10 code prediction is a multiclass, multilabel problem, each patient can have multiple diagnoses codes. Our model predicted codes also have multiple ICD-10 codes.

We evaluated ICD-10 code mapping using multilabel F1 scores, balancing precision and recall. Unlike traditional one-to-one classification, our setting reflects real-world clinical data, where each patient can have multiple ICD-10 codes. A match was counted if one or more predicted code matched a ground truth for each patient. Using exact code matches, the F1 was 0.245; when rolling codes up to ICD-10 category parents, F1 improved to 0.374, capturing broader diagnostic relevance. We also used weighted metrics to address class imbalance, ensuring that common conditions contributed proportionally to overall performance.

These F1 scores reflect the model's ability to correctly identify relevant diagnoses (minimizing false negatives) while avoiding incorrect ones (minimizing false positives), providing a clinically meaningful measure of overall diagnostic alignment.

### F. Distribution of Matching ICD-10 Diagnoses Across Primary, Secondary, and Tertiary Categories

In MIMIC-IV, each diagnosis is assigned a sequence number that reflects its clinical priority: 1 indicates the primary diagnosis, 2 the secondary, and 3 or higher represents tertiary or additional diagnoses.

To better understand how well our model predicts clinically important diagnoses, we analyzed the 59,742 exact ICD-10 code matches between our model and the MIMIC-IV ground truth. Given the many-to-many nature of diagnosis prediction, where each patient can have multiple predicted and ground truth codes, we selected the first available sequence number and its corresponding ICD-10 code per patient to categorize the match by priority.

Among the exact matches:

- Primary diagnosis (sequence number = 1): 41.1% of matches
- Secondary diagnosis (sequence number = 2): 17.1% of matches
- Tertiary or beyond (sequence number ≥ 3): 41.8% of matches

This indicates that while the model most frequently identifies the primary diagnosis, a large portion of correct predictions also capture secondary and tertiary conditions. This suggests the model's ability to extract a broad spectrum of relevant clinical information from unstructured notes. Figure. 6 visualizes the match distribution by sequence number, and Table 1 reports the exact proportions.

### DISCUSSION

In this study, we developed and evaluated a scalable NLP pipeline using scispaCy to extract diagnoses from MIMIC-IV discharge summaries and map them to standardized terminologies such as ICD-10 and SNOMED-CT. Beyond conventional entity recognition, our approach enables large-scale mapping of free-text clinical diagnoses and includes a task-specific pipeline to handle documentation challenges. To our knowledge, this is the first study to both build a scispaCy-based

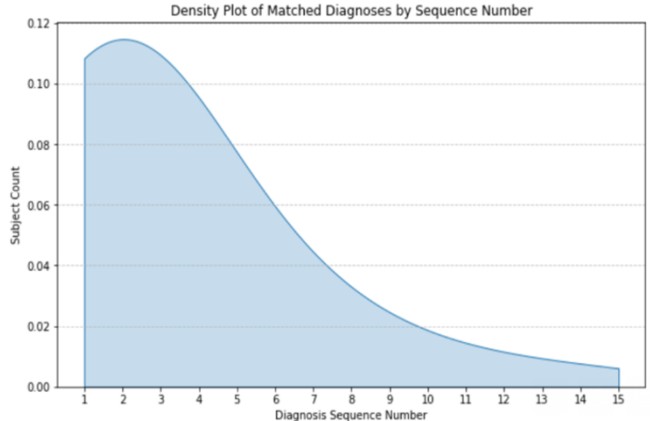

Fig. 6. Density Plot of Exact ICD-10 Code Matches by First Available Diagnosis Sequence Number in Ground Truth

TABLE I
PROPORTION OF PATIENTS WITH EXACT ICD-10 CODE MATCHES BASED ON FIRST AVAILABLE DIAGNOSIS SEQUENCE NUMBER IN GROUND TRUTH

| Sequence Number | Patient Statistics | | |
|---|---|---|---|
| | Total Patients | Matched Patients | Proportion |
| 1 | 59742 | 24574 | 0.411 |
| 2 | 59742 | 10249 | 0.171 |
| 3 | 59742 | 6649 | 0.111 |
| 4 | 59742 | 4519 | 0.075 |
| 5 | 59742 | 3282 | 0.054 |
| 6 | 59742 | 2422 | 0.040 |
| 7 | 59742 | 1810 | 0.030 |
| 8 | 59742 | 1417 | 0.023 |
| 9 | 59742 | 1111 | 0.018 |
| 10 | 59742 | 892 | 0.013 |
| 11 | 59742 | 608 | 0.010 |
| 12 | 59742 | 490 | 0.008 |
| 13 | 59742 | 401 | 0.006 |
| 14 | 59742 | 293 | 0.004 |
| 15 | 59742 | 1088 | 0.018 |

pipeline for mapping free-text clinical diagnoses and systematically compare the mapped outputs to structured MIMIC-IV diagnosis codes. All notes were de-identified per HIPAA standards; real-time EHR applications would require additional de-identification to ensure patient privacy compliance.

### Benchmarking Accuracy and Clinical Relevance

Our pipeline demonstrated robust coverage, successfully mapping 94.1% of extracted diagnosis concepts to UMLS CUIs, spanning 98% of patients in the cohort, with 80.1% of patients subsequently mapped to ICD-10 codes. Comparing to the MIMIC-IV structured 'Diagnosis' table, which primarily contains ICD-10 codes, 58.3% of patients had at least one exact ICD-10 code match between the model output and ground truth, reflecting the inherent complexity and granularity of medical coding [26], where even human coders can vary in specificity. A preliminary review of the 41.7% unmatched cases suggests broad reasons for mismatch, including non-disease entities (e.g., secondary diagnoses, social history),

contextual descriptions (e.g., syncope secondary to hypovolemia), synonym or phrasing variation (e.g., COPD exacerbation) and granularity differences. A more in-depth error analysis, including clinical expert review, will be conducted in future work. To further investigate granularity differences, we performed a roll-up comparison, where both generated and ground truth codes were rolled up to their category level parent codes within the ICD-10 hierarchy. This approach yielded a higher match rate of 83.1% of patients, reflecting the clinical relevance of capturing broader diagnostic categories even if the exact subcode is missed. This dual evaluation balances strict precision with meaningful clinical context, acknowledging that perfection in coding may not be the primary goal of automated tools, designed to aid rather than replace human expertise.

Importantly, we benchmarked our results against a recent study that evaluated the out-of-the-box performance of several large language models (including ChatGPT-4, Gemini 1.5, LLaMA 1.3, DeepSeek R1, and OpenAI's O3) on ICD coding tasks. That study found substantially lower accuracy, with the best model (OpenAI O3) reaching only 45.3% exact-code accuracy and an F1 score of 0.12 [25]. In contrast, our scispaCy-based pipeline achieved 58.3% exact-code accuracy (83.1% at the category level). This direct benchmark underscores that a lightweight, transparent, and computationally efficient pipeline can not only match but surpass the performance of current general-purpose LLMs for automated clinical coding.

A notable observation is that, while 41.1% of matched patients had a correct prediction for their primary diagnosis, this accounts for only approximately 24% of the full cohort of 102,539 patients with ground truth data, showing the challenge of accurately predicting the 'primary' diagnosis despite the overall accuracy.

### The Ground Truth Dilemma: Limitations of Structured Diagnoses

It is also critical to consider that the MIMIC-IV Diagnosis table serves as the ground truth in this study; however, the absence of raw strings from structured diagnosis data limits our ability to verify the validity of these mapped codes in the ground truth. Therefore, some NLP-identified codes missing from the ground truth may reflect true clinical findings rather than errors. This limitation likely contributes to the observed low F1 scores, 0.245 for exact matches and 0.374 for the category roll-up approach, as discrepancies may reflect genuine clinical information not captured in the ground truth dataset rather than just false positives or false negatives. Additionally, our evaluation is limited to a single institution's data, which may impact generalization. To improve these limitations of our study, future work will focus on expanding to multiple institutional datasets and those that contain raw clinical strings alongside human-coded mappings in ground truth, to enable more comprehensive validation. We also note that this study did not include validation by clinical experts, which limits our ability to fully assess the clinical correctness of the model-generated codes.

### Built for Deployment: Efficient, Scalable, Transparent, and Flexible

Despite some limitations, scispaCy offers practical advantages for clinical use. By leveraging UMLS CUIs, it flexibly maps to standard terminologies like ICD-10 and SNOMED-CT, with high entity match rates supporting research and decision-making. Its modular, rule- and statistics-based design runs on standard hardware(16 GB RAM, 4-core CPU), scales efficiently (processing over one million diagnoses from 100,000+ notes in 20 minutes), and can integrate with EHRs via standard APIs. Unlike black-box deep learning models, it ensures transparency and can be extended through APIs or LLM integration via the Model Context Protocol (MCP).

While MIMIC-IV contains unstructured notes with labeled section headers, many real-world clinical notes may lack such structure or follow different documentation practices, including variations in formatting, section naming, abbreviations, and phrasing. Our pipeline is flexible and does not rely on structured formatting. If headers are present, section-based filtering can be used or customized; if absent, the pipeline proceeds directly, as scispaCy maps clinical entities to UMLS CUIs, and a single line of code filters for the semantic type 'Disease or Syndrome' to isolate diagnosis-relevant concepts. These entities can then be mapped to standardized codes such as ICD-10 or SNOMED, enabling robust extraction even from fully unstructured notes. At the same time, national interoperability efforts, such as those led by the Office of the National Coordinator for Health IT (ONC) and the adoption of HL7 FHIR standards, continue to encourage more consistent structuring of EHR unstructured documentation [27]. Increased use of standardized templates and controlled vocabularies could reduce variability in clinical notes, complementing NLP-based approaches and further improving the reliability of automated diagnosis extraction.

### scispaCy as a Safer Alternative to LLMs for semantic mapping

Ethically and practically, our approach offers advantages over black-box LLMs. Unlike LLMs, which can sometimes hallucinate or generate inaccurate information [28,29], the scispaCy-based NLP pipeline uses rule-based mappings to UMLS CUIs, minimizing such risks. This approach ensures that extracted clinical concepts are reliable and grounded in established medical vocabularies. The deterministic nature of scispaCy provides consistent and interpretable results, making it well-suited for clinical applications requiring high accuracy and traceability. The risk of perpetuating racial or gender biases is relatively low in scispaCy, as it is trained on the UMLS Metathesaurus, a structured and curated vocabulary, unlike LLMs. While LLMs offer broader conceptual understanding, they carry a higher risk of reinforcing biases and often require additional mitigation tools [29].

### CONCLUSION

In a fragmented U.S. healthcare system, where interoperability remains challenging despite initiatives like the 21st Century Cures Act, our lightweight, transparent, and adaptable

pipeline provides a practical path for standardizing clinical data. It efficiently maps large volumes of unstructured notes to standard terminologies, supporting smaller organizations that lack resources for complex AI solutions. Dual-level evaluation, exact code matching and category roll-up, highlights the complexity of clinical coding. Compared to large language models, scispaCy offers a more controlled, interpretable, and less biased alternative. Future work should integrate richer datasets and explore human-in-the-loop frameworks to improve accuracy and clinical utility.

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

## DISCLOSURE

PN discloses that this study was conducted independently and is not related to her work at ConcertAI, where she is employed. She also notes that her participation in this study is carried out on a personal basis and is not funded. This applies solely to PN.
