# OpenReview forum: "Mapping Extracted Free-Text Primary Diagnoses to ICD-10 and SNOMED-CT Using SciSpacy: A Performance Evaluation"
_IEEE.org/EMBS/BHI/2025/Conference — BHI 2025_

### Official Review · Reviewer_cuJL · 2025-07-03
**Evaluating performance of scispaCy for mapping diagnosis concepts extracted from clinical EHR**

**Confidence:** 5
**Clarity Of Writing:** excellent
**Clinical Significance:** excellent
**Methodological Novelty:** excellent
**Overall Rating:** 7

**Experiments And Results:**

great

**Questions For The Authors:**

1. Have you conducted any qualitative or expert review of the model’s false positives and false negatives to better understand error modes?
2. How would the performance generalize to EHR datasets outside of MIMIC-IV or to less formal clinical documentation (e.g., outpatient notes)?

**Strengths:**

1. The work addresses a critical challenge in healthcare informatics, standardizing unstructured diagnoses for improved interoperability and decision support.
2. The pipeline runs efficiently on standard hardware and processes over 1 million diagnosis entries, enabling real-world applicability.

**Summary Of The Paper:**

This paper presents a scalable NLP pipeline leveraging scispaCy to extract free-text primary diagnoses from MIMIC-IV discharge summaries and map them to ICD-10 and SNOMED-CT codes. The pipeline utilizes UMLS CUIs as an intermediary step to crosswalk between free-text and standardized terminologies. Evaluation against MIMIC-IV ground truth shows a 94.1% CUI mapping rate, 80.3% mapping to ICD-10, and 58.3% exact ICD-10 code matches.

**Weaknesses:**

1. Despite high coverage, the F1 score (0.245) and even the roll-up score (0.374) remain low. While the authors contextualize this well, further performance enhancement may be necessary.
2. Generalizability remains unclear, as the evaluation is limited to MIMIC-IV.
3. The structured MIMIC-IV diagnosis table lacks raw diagnosis strings, complicating validation of false positives/negatives.

---

### Official Review · Reviewer_U9Eh · 2025-07-17
**Systematic evaluation of scispaCy for clinical diagnosis mapping shows practical promise but requires comparative analysis and enhanced validation**

**Confidence:** 2
**Clarity Of Writing:** good
**Clinical Significance:** good
**Methodological Novelty:** fair
**Overall Rating:** 5
**Final Rating:** 6

**Experiments And Results:**

fair

**Questions For The Authors:**

1. Baseline Performance: How does this approach compare to other automated clinical coding methods or human inter-coder agreement rates? What constitutes acceptable performance in automated clinical coding?
2. Error Pattern Analysis: Can you provide a detailed breakdown of failure modes for the 41.7% of patients without exact matches? Are there specific types of diagnoses or clinical scenarios where the system consistently underperforms?
3. Clinical Expert Validation: Have clinical coding experts evaluated the accuracy and appropriateness of the generated mappings beyond statistical comparison to structured codes?
4. Cross-Institutional Generalization: How would you expect performance to vary across different healthcare systems with varying documentation styles and coding practices?
5. Computational Efficiency: What are the processing times and resource requirements for the pipeline? How does this compare to manual coding workflows?
6. Improvement Strategies: Based on your analysis, what are the most promising approaches for improving the exact matching performance while maintaining the current hierarchical matching success?

**Strengths:**

1. Comprehensive Large-Scale Evaluation: The study demonstrates impressive empirical scope, processing over 1 million diagnosis entries from 145,000+ patients, providing substantial evidence for the approach's scalability and robustness.
2. Clinically Meaningful Evaluation Framework: The dual evaluation strategy (exact matching vs. hierarchical roll-up) thoughtfully acknowledges the complexity of medical coding practices. The 83.1% hierarchical matching rate represents clinically meaningful performance, recognizing that capturing correct diagnostic categories can be as valuable as exact code precision.
3. Practical Implementation Value: The pipeline offers significant practical advantages including transparency, standard hardware requirements, deterministic behavior, and lower bias risk compared to large language models. These features make it well-suited for clinical deployment.
4. First Systematic Assessment: This represents the first comprehensive evaluation of scispaCy for diagnosis code mapping at this scale, filling an important gap in understanding the practical utility of widely-used biomedical NLP tools.
5. High Entity Recognition Performance: The 94.1% success rate in mapping diagnosis concepts to UMLS CUIs demonstrates robust named entity recognition capabilities for clinical text.
6. Clear Methodology and Reproducibility: The authors provide detailed methodology and make code publicly available, supporting reproducibility and practical adoption.

**Summary Of The Paper:**

This paper presents a systematic evaluation of scispaCy for automated clinical diagnosis mapping using MIMIC-IV data. The authors develop a two-stage NLP pipeline: first extracting diagnosis statements from discharge summaries using rule-based methods, then employing scispaCy to map extracted concepts to UMLS CUIs and subsequently to ICD-10 and SNOMED-CT codes. Applied to 145,219 patients yielding over 1 million diagnosis concepts, the pipeline achieved 94.1% UMLS CUI mapping coverage across 98% of patients. When compared against MIMIC-IV ground truth, 58.3% of patients had exact ICD-10 code matches and 83.1% had hierarchical category-level matches. The authors position this as the first large-scale systematic evaluation of scispaCy for clinical diagnosis code mapping.

**Weaknesses:**

1. Absence of Comparative Evaluation: The paper's most significant limitation is the lack of comparison to other automated clinical coding approaches, human coder agreement rates, or established baselines. Without this context, it's impossible to assess whether the reported performance represents good, average, or poor results for automated clinical coding.
2. Ground Truth Validation Concerns: The reliance on MIMIC-IV structured diagnosis codes as ground truth is problematic, as acknowledged by the authors. The absence of raw clinical text validation limits confidence in the evaluation. The manual validation of only 200 diagnosis statements is insufficient for robust validation of the extraction process.
3. Limited F1 Performance: While the hierarchical matching shows promise, the exact matching F1 score of 0.245 and hierarchical F1 of 0.374 suggest substantial room for improvement, though clinical context for these metrics is lacking.
4. Single Institution Limitation: Using only MIMIC-IV data limits generalizability, particularly given known variations in clinical documentation practices across healthcare institutions.
5. Insufficient Error Analysis: The paper lacks detailed characterization of the 41.7% of patients without exact ICD-10 matches, missing opportunities to identify systematic failure modes and improvement strategies.
6. Methodological Limitations: The hierarchical mapping strategy prioritizing ICD-10 over SNOMED-CT lacks thorough justification beyond MIMIC-IV code prevalence. Additionally, the restriction to discharge summaries may not capture the full spectrum of clinical documentation.

---

### Official Review · Reviewer_DtqJ · 2025-07-18
**A valuable contribution to automation in clinical coding**

**Confidence:** 3
**Clarity Of Writing:** great
**Clinical Significance:** great
**Methodological Novelty:** good
**Overall Rating:** 7

**Experiments And Results:**

great

**Questions For The Authors:**

If this pipeline were to be deployed, do you expect unstructured medical records to have a section annotated explicitly as ‘primary diagnosis’ or similar? How might future work address this?

Could you clarify the key message intended by Figure 6 and Table 1, and how that fits into the rest of the narrative?

**Strengths:**

This manuscript fills an important gap in an age of LLMs, by providing baseline performance metrics for a straightforward and intuitive statistical NLP and rule-based pipeline. Documenting the performance of a straightforward NLP approach on MIMIC-IV addresses an important gap in the literature and provides an important comparison point for future research in medical coding to reference. The pipeline performs decently well given its simplicity, emphasizing that in many (most) cases a simple, resource-conscious approach to medical coding automation may still be competitive.

The manuscript was clear and well-written.

The practicality and potential for implementation in diverse settings is high.

To support trust, heuristics for data cleaning were manually validated on a small, random sample.

**Summary Of The Paper:**

This study proposes an automated framework for diagnostic code prediction from semi-structured medical notes and demonstrates its performance on the MIMIC-IV dataset. The proposed pipeline first selects the relevant sub-section for the diagnosis and cleans the data using rule-based heuristics. Next, it utilizes a pre-trained statistical NLP model from scispaCy for named entity recognition followed by mapping to the unified medical language system. Finally, a programmatic lookup leverages the structured nature of the unified medical language system to match entities to ICD-10  (or occasionally SNOMED) codes. The results showed that 58% of patients were matched to the exact ground truth ICD-10, with over 80% mapped to a higher level umbrella category of the ICD-10 ontology.

**Weaknesses:**

The selection of the ‘primary diagnosis’ sub-section of the MIMIC record seems to be a notable limitation of the work, given that it drops 75,000 patients which is not an insignificant number in this dataset. The number of patients lost in this step should be included in one of the flowchart figures (e.g. in Figure 3 or possibly Figure 4) for clarity and transparency.

One of the stated goals of the manuscript is to systematically evaluate the performance of the proposed pipeline. It would be beneficial to further explore the failure modes of this NLP system. If possible, I would like to see a more detailed breakdown of the 1.9% of records not assigned a CUI, the about 20% of records not mapped at a rolled-up level, and/or of the about 40% not mapped at an exact level. Even exploring a manual random sub-sample could be valuable here. Better understanding places where the pipeline breaks down would be scientifically valuable.

The paper mentions the trust- and explainability- related limitations of deep learning and generative AI methods. Despite these limitations, it would be helpful to provide an idea of the performance of these models, at least in the background section for context, even if the aim is not to compare to the current pipeline directly.

The last sentence of the abstract is misleading because human-in-the-loop is not demonstrated within this study.

---

### Official Review · Reviewer_UM46 · 2025-07-21
**Mapping Extracted Free-Text Primary Diagnoses to ICD-10 and SNOMED-CT Using SciSpacy: A Performance Evaluation**

**Confidence:** 3
**Clarity Of Writing:** good
**Clinical Significance:** good
**Methodological Novelty:** good
**Overall Rating:** 5
**Final Rating:** 6

**Experiments And Results:**

good

**Questions For The Authors:**

1) In the results, the authors mention that 83% (roll up) or 58% (exact) of patients had at least one model-generated ICD-10 code that matched the ground truth. My understanding is that Subsection F analyzes the distribution of predictions among the primary, secondary, and tertiary diagnoses within the matched patients who had at least one correct model-generated ICD-10 code. Could the authors please confirm if this interpretation is accurate?

2) It appears that many patients had multiple diagnoses. Since the proportions in Table 1 sum to 1, I assume the model generates only a single ICD-10 code per patient. If this is true, could the authors clarify why the model is limited to one prediction, and whether allowing multiple predictions per patient was considered?

**Strengths:**

1) I appreciate that the authors tackled the important problem of translating medical encodings using a rule-based approach rather than relying on LLMs. Their NLP pipeline appears well-suited for this specific task.

2) I think the evaluation is well thought out, I like that the authors included the roll-up comparison.

**Summary Of The Paper:**

The authors utilize SciSpacy to map diagnoses from UMLS CUI to ICD-10 and SNOMED-CT codes. They test their NLP pipeline using MIMIC-IV dataset. They use a rule-based script to extract relevant information from the discharge summaries and then map the diagnoses to UMLS using SciSpacy. Finally, the UMLS CUIs are crosswalked to ICD-10 and SNOMED-CT. They evaluate the performance of their NLP pipeline using an exact code comparison, which examined whether the generated ICD-10 code matched the true ICD-10 code, and a roll-up comparison, which compares the parent codes between the generated and true codes. They were able to obtain an F1 score of 0.245 for the exact match and a 0.374 for the roll-up comparison. And they were able to map 94% of the extracted diagnoses.

**Weaknesses:**

1) The authors draw a distinction between their rule-based NLP method and large language models (LLMs), raising valid concerns about the potential pitfalls of using LLMs for this task. However, the reported F1 scores suggest that the pipeline’s performance is relatively weak. Given these results, it would be helpful to see a direct comparison with an LLM to better understand the trade-offs. Additionally, the roll-up comparison reveals that the SciSpacy model often fails to predict diagnoses within the same parent code as the actual diagnosis, suggesting that its predictions may not be closely aligned with the true clinical labels.

2) Subsection F of the results is somewhat unclear.